# Peritumoral Edema in Gliomas: A Review of Mechanisms and Management

**DOI:** 10.3390/biomedicines11102731

**Published:** 2023-10-09

**Authors:** Kazufumi Ohmura, Hiroyuki Tomita, Akira Hara

**Affiliations:** 1Department of Tumor Pathology, Graduate School of Medicine, Gifu University, Gifu 501-1194, Japan; hinatavocco.0128@gmail.com (K.O.);; 2Department of Neurosurgery, Graduate School of Medicine, Gifu University, Gifu 501-1194, Japan; 3Center for One Medicine Innovative Translational Research, Institute for Advanced Study, Gifu University, Gifu 501-1193, Japan

**Keywords:** aquaporin, blood–brain barrier, edema, glioma, vascular endothelial growth factor

## Abstract

Treating malignant glioma is challenging owing to its highly invasive potential in healthy brain tissue and the formation of intense surrounding edema. Peritumoral edema in gliomas can lead to severe symptoms including neurological dysfunction and brain herniation. For the past 50 years, the standard treatment for peritumoral edema has been steroid therapy. However, the discovery of cerebral lymphatic vessels a decade ago prompted a re-evaluation of the mechanisms involved in brain fluid regulation and the formation of cerebral edema. This review aimed to describe the clinical features of peritumoral edema in gliomas. The mechanisms currently known to cause glioma-related edema are summarized, the limitations in current cerebral edema therapies are discussed, and the prospects for future cerebral edema therapies are presented. Further research concerning edema surrounding gliomas is needed to enhance patient prognosis and improve treatment efficacy.

## 1. Introduction

Cerebral edema is a clinical condition that occurs in various brain diseases and is characterized as an increase in the amount of fluid in the brain [1]. Head trauma, stroke, and brain tumors can cause cerebral edema [2]. Cerebral edema is generally classified into the following four types: (i) vasogenic edema owing to damage to the blood–brain barrier (BBB); (ii) cytotoxic edema resulting from the dysfunction of cell membranes; (iii) interstitial edema owing to the accumulation of cerebrospinal fluid (CSF) from within the ventricles into the extracellular space; and (iv) osmotic edema resulting from an imbalance of osmolality between the plasma and brain parenchyma [2,3]. Osmotic edema is a form of water intoxication or plasma hypoosmolarity. Vasogenic and cytotoxic edema are not always clearly distinguished clinically. These edemas tend to co-exist because of the symbiotic dependence of an intact BBB with adequate cerebral perfusion and a healthy astroglial system. For example, cerebral edema due to ischemic stroke is primarily caused by cytotoxic edema, which is then followed and exacerbated by vasogenic edema. Vasogenic edema can develop over days or weeks owing to ischemic damage to the BBB. Unlike other organs, the brain exists in a closed space owing to the presence of CSF and the skull; therefore, it is important to consider increased fluid in the brain as a pathological condition [4,5].

In 2021, the World Health Organization (WHO) updated its classification of central nervous system (CNS) tumors to prioritize molecular characteristics. The new classification for adult-type diffuse gliomas depends on isocitrate dehydrogenase (IDH) mutation and 1p/19q codeletion status, resulting in three categories. Notably, IDH-mutant astrocytoma with a homozygous deletion of cyclin-dependent kinase inhibitor 2A/B is now classified as WHO CNS grade 4, even without typical malignant glioma features. This revision also introduces specific WHO CNS grades 1–4 to distinguish CNS grading from systemic neoplasms [6]. Among brain tumors, malignant gliomas are known to have a poor prognosis despite progress in modern medicine [7,8]. Malignant gliomas have a high invasive potential, resulting in extensive edema formation surrounding the tumor [9]. Peritumoral edema is considered one of the main biological manifestations of glioma. Peritumoral edema can be evaluated preoperatively using neuroimaging. Glioma-related edema presents with many histopathological features, including infiltrating tumor cells, reactive astrocytes, activated microglia, and angiogenesis with different immunophenotypes. Indeed, the aggressiveness of glioma correlates more with the volume of edema than with its size, and WHO classifications of glioma are linked to tumor-related edema [1,3]. Peritumoral edema surrounding gliomas promotes increased intracranial pressure (ICP) and is associated with the development of neurological symptoms and brain herniation [10]. The degree of glioma-related edema is closely associated with recurrence and a poor prognosis. Furthermore, peritumoral edema is suspected to promote neoplastic glial cell invasion. Consequently, surgery and radio-chemotherapy aim to control both the tumor and the associated brain edema [1,9]. However, the mechanisms underlying the development of brain edema in gliomas remain unclear [2].

Steroids are commonly used to treat cerebral edema; however, their side effects and limited efficacy remain concerning [3], and the development of new therapies to treat cerebral edema is urgently needed. This review aims to describe the clinical features of peritumoral edema in gliomas. The mechanisms currently known to cause glioma-related edema are summarized, the limitations in current cerebral edema therapies are discussed, and the prospects for future cerebral edema therapies are presented.

## 2. Clinical Features of Glioma-Related Edema

### 2.1. Clinical Importance

Peritumoral edema indicates increased extracellular fluid, primarily in the cerebral white matter [11,12]. This edema causes symptoms such as headaches and vomiting. In addition, neurological dysfunction such as paralysis, language impairment, and visual impairment may occur. As glioma-related edema progresses, ICP increases, resulting in brain herniation, which can eventually lead to fatal consequences such as loss of consciousness and death [13,14]. Glioma-related edema is an indicator of tumor spread [11]. Extensive edema suggests that the glioma has invaded the surrounding healthy brain tissue [11,12]. Previous studies have reported that edema surrounding gliomas promotes glioma invasion and serves as an important predictor of tumor grade [15], recurrence [11,16,17], and poor prognosis [18]. Assessing the extent and progression of brain edema is crucial for treatment planning. Surgery, radiation therapy, and chemotherapy aim to control both the tumor and the surrounding edema [9,19]. Reducing edema is expected to prolong patient survival and enhance patient quality of life [13,14]. Edema is also a risk factor for complications during the treatment of brain tumors [1,3]. Increased ICP due to edema around the tumor can interfere with the blood supply and oxygenation to healthy brain tissue, leading to serious complications. Appropriate edema management is vital to prevent complications and ensure patient safety. Assessing peritumoral edema enables the selection of appropriate therapies and the monitoring of their effectiveness [11,16,19]. During post-treatment follow-up, changes in edema can be observed and used to assess tumor recurrence and treatment efficacy [16,20]. Thus, edema surrounding gliomas is clinically significant in pathological evaluation, treatment planning, prognosis prediction, and treatment efficacy monitoring.

### 2.2. Intracranial Pathophysiology of the Pressure–Volume Relationship and Cerebral Blood Flow (CBF) in Peritumoral Edema

Peritumoral cerebral edema adds to the mass effect of the tumor itself, disrupts tissue homeostasis, and reduces local blood flow, causing neurological disturbances. Components within the skull include the brain tissue, CSF, and intracranial blood volume, all of which must remain at a constant level in relation to each other. According to the Monro–Kellie law [1,2,4], any increase in the volume of one component must be compensated by a decrease in the volume of another component to maintain a stable ICP. However, this compensation reaches its limits when a maximal compression of the other components is reached leading to decompensation and increased ICP. Elevated ICP can negatively affect CBF. The brain requires a constant supply of oxygen and nutrients, and any reduction in CBF can lead to ischemia and tissue damage. When ICP becomes significantly elevated, it can exceed the mean arterial pressure (MAP), leading to a decrease in cerebral perfusion pressure (CPP). CPP is the pressure difference between MAP and ICP and is a critical determinant of CBF. CPP represents the pressure gradient that drives blood into the brain. A CPP below a certain threshold can lead to inadequate CBF, potentially causing brain ischemia [1,2,4].

### 2.3. Imaging Findings

Edema surrounding gliomas can be visualized using imaging modalities such as computed tomography (CT) and magnetic resonance imaging (MRI). Using MRI, edema surrounding a tumor appears as a region of high intensity on T2-weighted images [20]. Neuroimaging plays a crucial role in evaluating the presence and extent of peritumoral edema [20,21]. However, malignant gliomas tend to invade healthy brain tissue, and differentiating between peritumoral edema and glioma invasion can be challenging, which is problematic given imaging findings affect treatment planning during surgical resection and radiation therapy [16,18]. The physiological basis of gadolinium contrast-enhanced MRI, a conventional MRI, is the destruction of the BBB, which reflects tumor invasion and angiogenesis in gliomas, but can also be attributed to various other factors, such as acute reactive changes after surgery or radiation, steroids, and radiation necrosis [20]. Notably, corticosteroids can change the appearance of a glioma on an MRI, which has effects on the size of the contrast-enhanced tumor and peritumoral edema. Ongoing corticosteroid therapy is a response assessment in neuro-oncology criterion used to evaluate treatment efficacy [20]. Determining disease progression can be complicated during steroid treatment. Therefore, studies have focused on developing neuroimaging methods for effective differentiation [21,22,23]. Previous studies have explored the use of magnetic resonance spectroscopy [24] and functional MRI [25] to differentiate brain edema from glioma invasion. Nuclear imaging approaches such as positron emission tomography have also been reported [26,27]. Recently, artificial intelligence has been used to facilitate the differentiation of gliomas from tumor invasions [28,29,30]. However, it should be noted that edema surrounding gliomas can act as a source of nutrients (known as a niche) and facilitate tumor invasion [9,19], which can make it challenging to differentiate between edema and tumor invasion using imaging alone.

## 3. Mechanisms Underlying the Formation of Glioma-Related Edema 

### 3.1. Fluid Regulation in the Brain

The brain contains >80% water, an amount that is proportionally higher than that found in other organs. The distribution of fluids in the brain can be classified into four categories: intracellular fluid (ICF), interstitial fluid (ISF), plasma, and CSF [2,9]. In total, 140 mL of CSF is produced in the subarachnoid space and ventricles [5]. Conversely, the brain parenchyma contains 280 mL of ISF, twice the volume of CSF [2,5]. The overall volume of these fluids, including the blood, remains constant within the intracranial tissue environment, as governed by the Monro–Kellie law [1,4,31]. Plasma and ISF are separated through the BBB, whereas plasma and CSF are separated through the blood–CSF barrier [2,5]. Given the exchange of fluids between the ISF and CSF, they are sometimes collectively referred to as neurofluids [32]. Similar to other organs, water influx into the brain is primarily driven by hydrostatic and colloid osmotic pressures, allowing water to exit the intracerebral microvascular network and ventricular choroid plexus and be subsequently absorbed by the tissues [33]. However, blood supply to the ventricular choroid plexus accounts for only 1% of the cerebral microvascular network [5], highlighting the microvascular network as the primary system responsible for water inflow and outflow in the brain (Figure 1). The primary function of the choroid plexuses is to produce CSF. However, it only represents <20% of the plasma flow to the choroid plexus [5]. As previously mentioned, blood flow to the ventricular choroid plexus constitutes only 1% of the cerebral microvascular network. Consequently, it is important to note that cerebral capillaries, including those in the choroid plexus, contribute to CSF production. This process involves capillaries generating ISF through hydrostatic pressure differences, which is subsequently absorbed by the capillaries because of osmotic pressure differences [2,5]. Notably, some of this ISF combines with CSF to contribute to the overall CSF volume within the ventricles and subarachnoid space.

Nevertheless, the mechanisms of fluid drainage in the brain remain unclear. Previously, the primary pathway for fluid efflux was considered to be ‘bulk flow’. Bulk flow involves the absorption of CSF from the arachnoid granules that then flows into the venous sinus [33,34]. However, based on recent studies, it is now considered that CSF absorption from arachnoid granules does not occur when ICP is normal, suggesting that CSF drainage may primarily occur when ICP is elevated [31,35,36]. In addition to bulk flow, several other drainage pathways have been proposed [36,37,38,39,40,41,42,43,44,45,46]. Herein, we outline three representative pathways (Figure 2). First, a major portion of fluid drains into the venous system via the BBB [5,33,37]. Water passes through the BBB as a result of hydrostatic and osmotic pressures. The endothelial glycocalyx, a delicate carbohydrate-rich layer enveloping the luminal surface of the endothelium, is the first component of the BBB, acting as an endothelial gatekeeper. The endothelial glycocalyx is denser in the brain than in other organs, suggesting a critical barrier function in the CNS. The glycocalyx is damaged in diseases such as ischemia and infection, and in trauma [2]. Hence, the glycocalyx is implicated in the pathogenesis of cerebral edema of various etiologies. Second, the glymphatic system involves fluid drainage through the basement membrane in the capillary walls; this fluid then progresses into the perivascular space to the cervical lymph nodes along with the blood flow [37,43,46]. Aquaporin 4, present in astrocyte endfeet around cerebral capillaries, assists in the excretion of fluid via this drainage route. Glymphatic drainage has diurnal variations, with flow highest during sleep. The final mechanism of fluid drainage in the brain is via intramural periarterial drainage, where fluid drains into the basement membrane surrounding the capillaries and flows in a retrograde manner, with blood flowing through the basement membrane of the smooth muscle in contiguous capillaries and cerebral arteries [38,44]. Ultimately, the fluid drains into the internal carotid artery and then into the deep cervical lymph vessels in the neck without entering the subarachnoid space. Disorder in this pathway has recently been considered a major etiology of cerebral amyloid angiopathy and has been studied intensively in association with Alzheimer’s disease [44]. It had long been considered that the CNS lacked lymphatic vessels [44], which, in intracranial tissues, implied that when plasma leaked into the ISF, it could not be processed, resulting in increased osmotic pressure and fluid accumulation. However, the discovery of cerebral lymphatic vessels in a 2015 study dramatically changed this understanding [43]. The latter two drainage pathways eventually connect to the deep cervical lymph vessels, suggesting that they may also function as lymphatic vessels similar to those found in other organs [44,46].

### 3.2. Mechanisms of Glioma-Related Edema Formation

Peritumoral edema formation surrounding gliomas is considered to be driven by three main mechanisms [47,48,49,50,51]: (i) increased BBB permeability, (ii) tumor angiogenesis, and (iii) increased aquaporin 4 expression (Figure 3). The BBB plays a central role in the development of peritumoral edema. The BBB, which comprises pericytes, endothelial cells, astrocytes, and a basement membrane, regulates vascular permeability and prevents plasma extravasation into the interstitial space of the brain [5,9,51]. As mentioned earlier, the BBB is also responsible for the drainage and influx of brain fluid, with an estimated daily balance of approximately 40,000 mol of water entering and exiting the brain [46]. However, in the presence of excess fluid, fluid pooling and subsequent edema occur. As gliomas grow, they secrete angiogenic factors such as vascular endothelial growth factor (VEGF) and matrix metalloproteinases (MMPs) [52,53,54,55]. These factors increase the vascular permeability of the BBB, allowing plasma to enter the extracellular space and leading to edema formation. The overexpression of cyclooxygenase-2 (COX-2) in gliomas is also associated with increased vascular permeability [56,57]. Other factors contributing to increased BBB permeability include leukotrienes, nitric oxide, and mast cell activation. 

The most important and most widely studied mediator is VEGF. VEGF binds to capillary endothelial cells via VEGF receptors. VEGF downregulates the expression and structure of tight junction proteins, resulting in increased vascular permeability through expanding the fenestra of endothelial cell clefts and segmental endothelial cells. The conditions within the tumors, such as hypoxia and acidosis, can further increase VEGF production, leading to widening of interendothelial gaps. Multiple oncogenes and tumor-suppressor genes (such as SRC, RAS, and TP53), hormones, cytokines, and various signaling molecules (including nitric oxide and mitogen-activated protein kinases) can also regulate VEGF expression [49,51]. VEGF has a dual effect in terms of increasing vascular permeability and tumor angiogenesis. Furthermore, it has been suggested that VEGF has a role in anti-tumor immunity. VEGF has received much attention within the tumor-related microenvironment.

There are approximately 20 types of MMPs, which are endopeptidases that cleave the protein of components of the extracellular matrix components. The extracellular matrix constitutes the basement membrane of the BBB and, under normal conditions, its function is regulated by MMPs. Excessive MMP activity leads to the cleavage of the basement membrane and disruption of the BBB. MMP2 and MMP9, expressed in the CNS, are activated during brain damage. Since MMP inhibitors suppress cerebral edema formation owing to vasogenic edema, excessive MMP activation is considered a factor in the development of glioma-related edema [52,54].

Tumor blood vessels formed under the influence of angiogenic factors do not possess normal BBB characteristics observed in healthy cerebral blood vessels [58,59,60,61]. These abnormal tumor vessels contribute to the formation of edema. Tumor vessels exhibit reduced expression of tight junction proteins, such as occludin, claudin, cadherin, and zona occludens 1 and 2, as well as functional adhesion molecules that maintain tight junctions in the healthy endothelium [60]. Consequently, the integrity of endothelial tight junctions is weakened, disrupting normal barrier function and enabling plasma to enter the extracellular space, leading to edema formation. Previous electron microscopy studies of tumor vessels in human glioblastomas have shown hyperplastic tumor endothelial cells with prominent surface infoldings, tortuous tight junctions between endothelial cells, and irregular basement membranes [49,59,61]. 

Aquaporins, a group of water channel proteins, have also been implicated in the development of peritumoral edema [50]. Aquaporins play a vital role in maintaining water homeostasis. Among 13 identified aquaporin types (aquaporin 0–12), aquaporins 1, 4, 5, and 9 are found in the CNS, with aquaporin 4 being particularly important for brain fluid regulation [50,62,63,64]. Aquaporin 4 is a highly water-selective transport channel. Aquaporin 4 passively transports water in response to the osmotic pressure difference between the interior and exterior of the membrane. Aquaporin 4 is expressed in astrocyte endfoot processes surrounding capillaries, in astrocyte processes comprising the glial limiting membrane, in ependymal cells, and in subependymal astrocytes. This distribution suggests that aquaporin 4 controls water fluxes into and out of the brain parenchyma [64,65,66]. Aquaporin 4 overexpression has been observed in gliomas and is associated with edema formation [66]. In gliomas, aquaporin 4 is highly concentrated at the cell membrane covering the surface of tumor cells and is strongly upregulated and redistributed across the surface of glioma cells. Furthermore, aquaporin 4 expression has been linked not only to edema formation in gliomas, but also to malignancy and VEGF expression [66]. In addition to osmotic pressure, hypoxia is an important factor affecting aquaporin 4 regulation in relation to gliomas. However, the precise mechanism through which aquaporin 4 contributes to edema formation remains unclear. Aquaporin 4 is also considered to be involved in fluid drainage, leading to complex edema formation [2,63].

Aquaporin 1 mediates osmotic and hydrostatic water fluxes across cell membranes. It is mainly found in the choroid plexus. Imbalances between CSF production and absorption can result in hydrocephalus. The upregulation of the water channel aquaporin 1 is a unique feature of glioblastoma, although aquaporin 1’s role in glioma has not been fully elucidated. In a preclinical study, aquaporin 1 blockers were shown to have the potential to reduce water fluxes across the choroid plexus and decrease ICP. Aquaporin 1 is upregulated in response to increased tumor glucose consumption [62]. Aquaporin 1 expression has been found to correlate with the aggressiveness of the tumor. In terms of the role of aquaporin 1 in glioblastoma, it has been suggested that the water channel provides for a regulated decrease in tumor volume, which facilitates increased invasion through the restricted extracellular space in the parenchyma [62,64]. The association between the other aquaporin groups and peritumoral edema remains unclear.

The effect of gliomas on fluid drainage has not been previously investigated. It is possible that the abnormal vascular structure of the tumor lacks a normal BBB [60,61], hindering normal fluid drainage. In other words, intramural periarterial drainage may be compromised owing to the absence of a typical basement membrane structure [38,44]. Gliomas have high invasive potential, particularly through perivascular invasion. Perivascular invasion owing to gliomas may inhibit perivascular drainage, such as through the glymphatic system, or intramural periarterial drainage [12,19], but the precise effects of these drainage pathways in gliomas remain unclear.

## 4. Prospects for Treatment of Cerebral Edema

The drugs used in clinical practice to treat peritumoral edema are summarized in Table 1. Figure 4 shows a simplified mechanism of action in the drugs used to treat glioma-related edema.

### 4.1. Osmotherapy

Osmotherapy involves the use of mannitol or hyperosmotic saline to rapidly reduce brain pressure in patients with increased ICP due to cerebral edema [1,3]. Mannitol and hyperosmotic saline increase intravascular colloid osmotic pressure, leading to the withdrawal of water from the brain tissue. The intravenous administration of these agents can effectively alleviate brain pressure within minutes to half an hour [1,67]. However, it is essential to note that osmotherapy is intended for emergency situations and that its effects are temporary, lasting only a few hours [1,23]. Long-term administration is not recommended because of the potential side effects, such as electrolyte imbalance and renal dysfunction [23]. Additionally, in cases of edema surrounding gliomas, mannitol may leak from vessels into the brain parenchyma, potentially worsening the edema [51,68].

### 4.2. Steroids

The use of steroids for managing edema around brain tumors dates back half a century. Since then, steroids have become the standard treatment for edema [69,70,71,72]. Among various steroids, dexamethasone is commonly prescribed because of its low mineralocorticoid content and prolonged effectiveness [69]. Although the precise mechanism of action of steroids in treating cerebral edema remains only partially understood, one hypothesis suggests that they exert their edema-improving effects through inhibiting tumor-secreted VEGF and reducing increased vascular permeability [73,74,75]. Additionally, steroids have been reported to inhibit cerebral edema through enhancing the expression of claudin, occludin, and cadherin, thereby strengthening the adhesion of vascular endothelial cells and preventing plasma leakage into surrounding tissues [70]. It is important to note that, despite their efficacy against cerebral edema, long-term steroid use poses risks of side effects. For example, prolonged use may increase the likelihood of osteoporosis and susceptibility to infection [69,76].

### 4.3. Anti-VEGF Agents

As mentioned previously, VEGF plays a critical role in tumor edema by increasing vascular permeability and promoting tumor angiogenesis [52,77]. VEGF binds to capillary endothelial cells through tyrosine kinase receptors and is estimated to be approximately 1000 times more vascularly permeable than histamine [78]. In gliomas, VEGF expression correlates with malignancy [48].

Bevacizumab, a monoclonal antibody targeting VEGF, and cediranib, a tyrosine kinase inhibitor of the VEGF receptor, have been reported to reduce peritumoral edema in clinical studies [79,80]. Notably, bevacizumab has received approval from the United States Food and Drug Administration for the treatment of recurrent glioblastomas [52,81]. These drugs, often referred to as ‘super steroids’, have shown marked improvements in treating edema [81,82]. However, it is essential to consider their high treatment costs and serious side effects, such as hemorrhage and embolism [83].

Furthermore, studies suggest that angiogenesis inhibitors for gliomas primarily suppress brain edema but may lead to increased tumor invasion [84]. This observation supports the hypothesis that tumor cell infiltration serves as a means of obtaining nutrition, with the surrounding edema providing nourishment to gliomas. Consequently, when edema is no longer a source of nutrition, tumors may seek alternative sources and potentially invade healthy tissues.

### 4.4. Development of New Therapeutic Agents

Considering the side effects and limited efficacy associated with steroids, it is imperative to explore and develop novel therapies for peritumoral edema. Various therapeutic approaches are currently being investigated, and some promising prospects are outlined below.

#### 4.4.1. COX-2 Inhibitors

COX-2, which is overexpressed in gliomas, leads to increased production of prostaglandin E2 [56,57,85]. This molecule has been linked to tumor angiogenesis owing to its vasodilatory effects and may contribute to edema formation [56]. COX-2 has garnered attention not only for its role in tumor angiogenesis but also in relation to its involvement in glioma invasion, proliferation, and development of treatment resistance [57].

Animal studies using celecoxib, a COX-2 inhibitor, have shown reduced VEGF expression, suggesting an anti-edema effect [86,87]. Despite their potential benefits, clinical studies of COX-2 inhibitors for glioblastoma have shown limited efficacy over the past decade [88,89,90]. COX-2 inhibitors are associated with certain adverse effects, such as myocardial infarction, stroke, and other cardiovascular disorders [91]. The mechanisms underlying these side effects remain unclear, which has led to the discontinuation of several previously available COX-2 inhibitors [85,91]. Careful evaluation of the safety profile is essential for the development of potential COX-2 inhibitor-based therapies for cerebral edema.

#### 4.4.2. Aquaporin Inhibitors

The specific mechanism through which aquaporins contribute to edema formation surrounding tumors remains unclear, necessitating further research to understand their role in humans [92,93,94,95]. Nevertheless, it is possible that regulating aquaporins could lead to improvements in peritumoral edema. Goreisan, a herbal medicine used in Japan and other Asian countries (known as Wu Ling San in China and Oreongsan in Korea), is considered to inhibit edema through downregulating aquaporin 4 expression [96,97,98,99]. Studies have demonstrated its efficacy in reducing cerebral edema following stroke [100]. Moreover, it has been shown to improve cerebral edema in animal studies [101,102]. While these findings are encouraging, more comprehensive studies evaluating their effect on peritumoral edema are urgently needed to establish their potential as therapeutic agents for peritumoral edema [103].

#### 4.4.3. Boswellic Acids

Boswellic acids are phytotherapeutic agents obtained from *Boswellia serrata. Boswellia serrata* is an Indian plant that produces frankincense and contains at least six different boswellic acids, including the main bioactive component acetyl-11-keto-b-boswellic acid (AKBA) [104]. AKBA has an anti-inflammatory function, which is known to play a positive role in brain edema [105,106,107]. Furthermore, AKBA plays a pharmacological role in relation to anti-infection, antitumor, anti-oxidant, and anti-aging processes [105]. There are two main pathways affecting peritumoral edema, namely, inhibition of VEGF expression and 5-lipoxygenase. AKBA is a potent inhibitor of hypoxia-inducible factor-1a, which results in the downstream inhibition of VEGF expression and reduction in brain edema. AKBA inhibits 5-lipoxygenase, which is a key enzyme in leukotriene synthesis from arachidonic acid. Leukotrienes have a strong influence on vascular permeability [105].

In one prospective trial, 44 patients with primary or secondary malignant cerebral tumors were randomly assigned to receive radiotherapy plus either boswellic acids or placebo. The area of brain edema, measured using MRI, showed a significant reduction of >75% in 60% of patients treated with boswellic acids compared with 26% of patients in the placebo group [108]. More recently, in a study of 20 patients with glioblastoma treated with boswellic acids while receiving radiation therapy, treatment with boswellic acids significantly reduced the radiochemotherapy-induced cerebral edema, with no severe adverse events observed. Only six patients had minor gastrointestinal discomfort. The supplement 5-Loxin, which provides 30% AKBA according to weight, was also reported to have positive effects on radiation-induced brain edema in a case series [109]. 

### 4.5. Future Perspectives

Effective treatment for peritumoral edema with fewer side effects and lower costs is needed. Ease of administration and high patient tolerability are also important. Steroids and anti-VEGF drugs have various side effects, which can markedly reduce patient quality of life. Anti-VEGF drugs require an implanted venous access device. Furthermore, given that bevacizumab injections cost between USD $5000–$7000 per injection, the issue of costs for patients needs to be taken into account [52]. Based on these factors, the use of anti-aquaporin 4 and boswellic acids in combination is worth considering, as both cost less and can be taken orally [99,104]. However, despite some reports on their efficacy, the lack of long-term treatment results or large-scale clinical trials in relation to peritumoral edema remains problematic. Larger studies are required to assess the potential steroid- or anti-VEGF-sparing effects of both drugs. Furthermore, a better understanding of the molecular mechanisms underlying the functional activity of aquaporin 4 and AKBA may help in exploiting their full therapeutic potential.

## 5. Conclusions

Peritumoral edema, an important characteristic in patients with gliomas, profoundly affects their neurological symptoms and prognosis. Although neuroimaging facilitates confirmation of brain edema, it is essential to differentiate brain edema from glioma infiltration. The edema that is observed surrounding gliomas is likely attributable to BBB permeability and tumor angiogenesis. Steroids serve as the primary treatment for peritumoral edema. However, their limited efficacy and potential side effects have raised concerns. Anti-VEGF agents have shown potential to reduce edema surrounding tumors, with several studies validating their efficacy; however, their high treatment costs and serious side effects, such as hemorrhage and embolism remain problematic.

New therapeutic strategies targeting aquaporin 4 appear to be promising. The utilization of boswellic acids may also be potentially useful. It is anticipated that future research on peritumoral edema will lead to discoveries that may ultimately enhance patient prognosis and therapeutic outcomes.

## Figures and Tables

**Figure 1 biomedicines-11-02731-f001:**
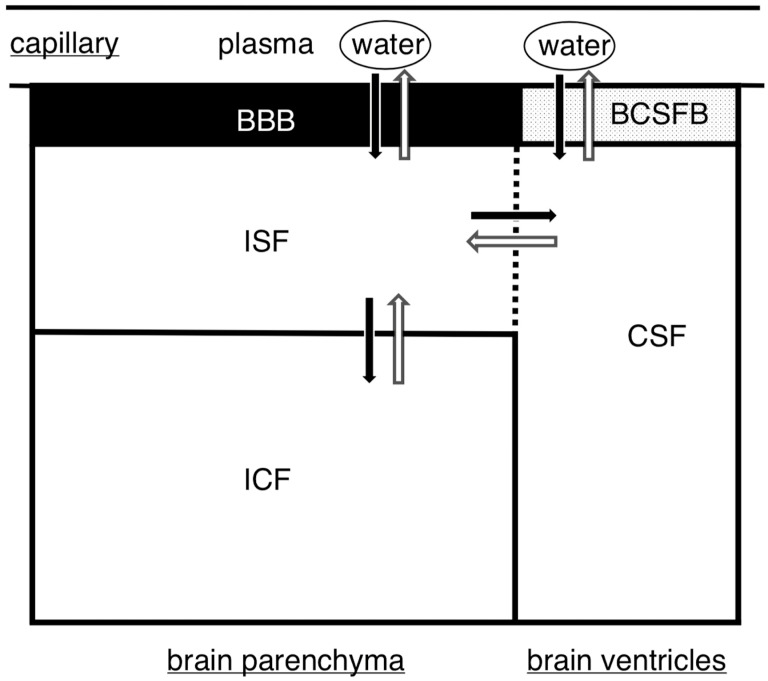
A schematic diagram of cerebral fluid compartments. The brain comprises four compartments. In addition to ICF and ISF, the presence of CSF is a defining feature in that CSF can move freely to and from the ISF. However, plasma is strictly separated by the BBB or by the blood–cerebrospinal fluid barrier. BBB, blood–brain barrier; BCSFB, blood–cerebrospinal fluid barrier; CSF, cerebrospinal fluid; ICF, intracellular fluid; ISF, interstitial fluid.

**Figure 2 biomedicines-11-02731-f002:**
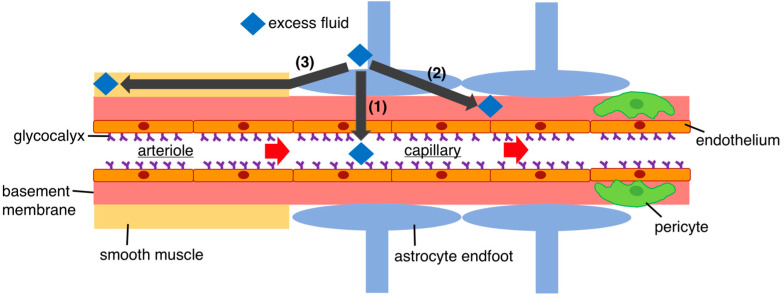
Pathways of excess fluid drainage. Several pathways for draining fluid from the brain have been identified, as follows. (1) BBB drainage: This is an important route for fluid drainage, and the BBB allows for the constant movement of water in and out, thereby regulating fluid balance in the brain. Water passes through the BBB as a result of hydrostatic and osmotic pressures. (2) Glymphatic drainage: This pathway involves the movement of fluid through the perivascular space in conjunction with blood flow; the fluid eventually drains into the subarachnoid space. Aquaporin 4 is present in astrocyte endfeet around the cerebral capillaries, which assist in the excretion of fluid via this drainage route. (3) Intramural periarterial drainage: In this process, fluid drains into the basement membrane surrounding the capillaries. Fluid then flows backward with the blood through the basement membrane of contiguous microvessels and cerebral arteries towards the surface of the brain. Subsequently, fluid drains into the internal carotid artery and eventually reaches the deep cervical lymphatic vessels in the neck, bypassing the subarachnoid space. Pathways (2) and (3) both ultimately connect to lymphatic vessels. Consequently, these pathways have recently been considered equivalent to lymphatic vessels found in other organs. Pathway (2) requires several hours to drain excess fluid, whereas pathway (3) can drain more quickly.

**Figure 3 biomedicines-11-02731-f003:**
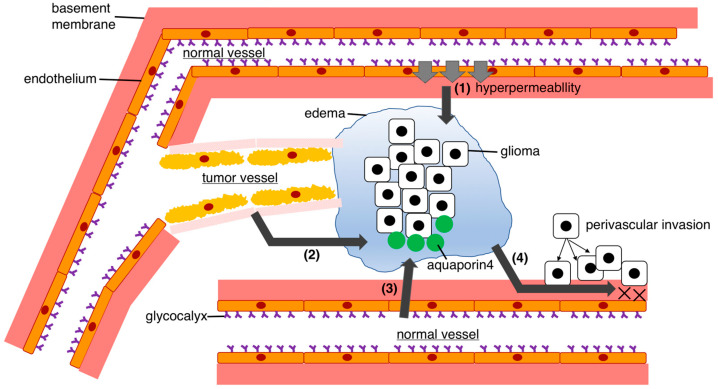
Mechanism of peri-glioma edema formation. Gliomas cause pronounced edema in surrounding regions. The mechanisms underlying this process involve the following steps: (1) The release of vascular permeability factors: vascular permeability factors such as VEGF and matrix metalloproteinases are released, leading to the accumulation of fluid that can easily traverse the BBB and enter the brain, causing fluid pooling. (2) Tumor angiogenesis: tumor angiogenesis occurs owing to tumor proliferation, resulting in abnormal tumor vessels without a functional BBB. Tumor vessels exhibit reduced expression of tight junction proteins and functional adhesion molecules. Consequently, it enables plasma to enter the extracellular space, leading to edema formation. (3) Overexpression of aquaporin 4: gliomas exhibit an increased expression of aquaporin 4, which contributes to fluid retention and subsequent edema. Aquaporin 4 controls water fluxes into and out of the brain parenchyma. However, the precise mechanism through which aquaporin 4 contributes to edema formation remains unclear. (4) Perivascular glioma invasion: perivascular invasion by gliomas may interfere with the brain’s normal fluid drainage pathways, such as the glymphatic system, or intramural periarterial drainage, which may potentially exacerbate edema formation. BBB, blood–brain barrier; VEGF, vascular endothelial growth factor.

**Figure 4 biomedicines-11-02731-f004:**
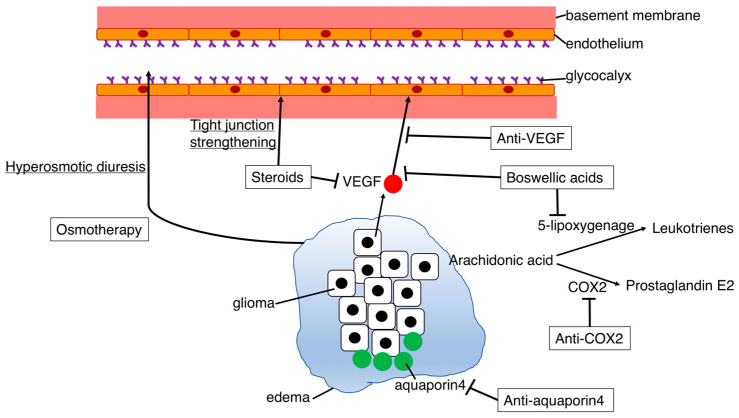
The mechanisms of action in drugs used to treat glioma-related edema. Several drugs for peritumoral edema have been used clinically, as follows. (1) Osmotherapy: Mannitol or hyperosmotic saline increases intravascular colloid osmotic pressure, leading to withdrawal of water from the brain tissue. Intravenous administration of these agents can effectively alleviate brain pressure within minutes to half an hour. However, it is essential to note that osmotherapy is intended for emergency situations and that its effects are temporary, lasting only a few hours. (2) Steroids: Steroids exert their edema-improving effects through inhibiting tumor-secreted VEGF and reducing increased vascular permeability. Additionally, steroids have been reported to inhibit cerebral edema through enhancing the expression of claudin, occludin, and cadherin, thereby strengthening the adhesion of vascular endothelial cells and preventing plasma leakage into surrounding tissues. While steroids remain the standard therapy for edema, various side effects are known to occur with long-term use. (3) Anti-VEGF: VEGF inhibitors suppress VEGF, playing a critical role in tumor edema through increasing vascular permeability and promoting tumor angiogenesis. These have strong antitumor edema effects, but side effects such as bleeding and embolism are concerning, as well as their high cost. (4) Anti-cyclooxygenase-2 (COX-2): COX-2 inhibitors suppress the synthesis of prostaglandin E2. Prostaglandin E2 has been suggested to be involved in tumor angiogenesis, as well as increased vascular permeability; however, cardiovascular disorders are a well-known side effect. (5) Anti-aquaporin 4: Overexpression of aquaporin 4 has been observed in gliomas and is associated with edema formation. Goreisan, a herbal medicine used in Japan and other Asian countries (known as Wu Ling San in China and Oreongsan in Korea), is considered to inhibit edema by downregulating aquaporin 4 expression. However, the precise mechanism through which aquaporin 4 contributes to edema formation remains unclear. (6) Boswellic acids: boswellic acids are phytotherapeutic agents obtained through extraction from *Boswellia serrata.* Boswellic acids mainly include the bioactive component acetyl-11-keto-b-boswellic acid (AKBA), which has an anti-inflammatory function, and are known to play a positive role in brain edema. The two main pathways affecting peritumoral edema involve the inhibition of VEGF expression and 5-lipoxygenase. AKBA inhibits 5-lipoxygenase, which is a key enzyme in leukotriene synthesis from arachidonic acid. Leukotrienes have a strong influence on vascular permeability. VEGF, vascular endothelial growth factor.

**Table 1 biomedicines-11-02731-t001:** List of drugs used to treat edema surrounding gliomas.

Drug (Example)	Mechanism	Edema-Improving Effect	Side Effects
Osmotherapy (mannitol)	Hyperosmotic diuresis	Immediate effect	A temporary effect only, rebound
Steroids (dexamethasone)	Tight junction strengthening and suppression of increased vascular permeability	Standard treatment	Infection, weight gain, many side effects
Anti-VEGF (bevacizumab)	Inhibits VEGF → suppresses vascular permeability and tumor angiogenesis	Strong effect	Hemorrhage, embolism, high drug cost
Anti-COX-2 (celecoxib)	Inhibition of prostaglandin production → suppresses increased vascular permeability	Unknown	Cardiovascular disorder
Anti-aquaporin 4 (goreisan)	Water accumulation control?	Unknown	Unknown
Boswellic acids (5-Loxin)	Interference with the VEGF pathway and leukotriene formation	Unknown	Gastrointestinal side effects

COX-2, cyclooxygenase-2; VEGF, vascular endothelial growth factor.

## Data Availability

No new data were created or analyzed in this study. Data sharing is not applicable to this article.

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
