# Peer review of "Peritumoral Edema in Gliomas: A Review of Mechanisms and Management"

_biomedicines, 2023, doi:10.3390/biomedicines11102731_

Round 1

Reviewer 1 Report

Ref: biomedicines-2560399
Title: Peritumoral edema in gliomas: a review of mechanisms and management
Journal: Biomedicines

The manuscript entitled: “Peritumoral edema in gliomas: a review of mechanisms and management” by Ohmura et al. is a well written article reviewing the clinical features and mechanisms of peritumoral edema in gliomas. The article fits within the scope of the journal and the special issue. I recommend this manuscript to be accepted for publication in the Biomedicine journal, after some major revisions (please see below):

 -Introduction: it would be good to be included the glioma grades according to the WHO 2021 classification and their association with glioma-related edema.

-Figure 2 legend: please expand on the legend by explaining the three routes indicating in the figure. The same for figure-3: figure legends need to be more explanatory.

-Lines 159-163: please expand and explain more the glioma-related edema mechanism; which MMPs members are involved? Which other proteins/pathways?

-Lines 175-176: please explain with more details how the overexpression of aquaporin 4 leads to edema. Are there other members of aquaporins that potentially may cause edema?

-Table 1: it would be good in the first column of the table (or as an additional column) to add a few examples of specific drug names that are currently used as well.

- Drugs used to treat edema surrounding gliomas: it would be useful to include a summative figure of the mechanisms of action of the most crucial drugs that are currently used-which pathways those drugs block etc.

-Comparative, from a critical point of view addition (1-2 paragraphs) of the drugs is recommended, as well as future perspectives.

Reviewer 2 Report

In the classification of edemas, it is omitted that combined forms of vasogenic and ischemic can develop in an advanced stage of oedema. Osmotic edema is a form of water intoxication or plasma hypoosmolarity.  

In „Clinical importance”, the description of the intracranial pathophysiology of pressure-volume relation and cerebral blood flow is imprecise; the perfusion pressure and development of ischemia are not clearly written.

The description of neuroimaging of edema is imprecise, sometimes misleading. For example, the authors write that " Previous studies have explored the use of magnetic resonance spectroscopy [23] and functional MRI [24] to differentiate brain edema from glioma invasion." The cited paper [24] does not deal with fMRI, not to mention that fMRI is not suitable for differential diagnosis of edema. The authors must discuss the importance of contrast enhancement in differentiating edema from tumor tissue. The authors do not discuss that edema and the effect of steroids on edema can be measured quantitatively by MRI.

The „Fluid regulation in the brain” section is problematic. The authors state,” However, blood supply to the ventricular cho- 103 roid plexus accounts for only 1% of the cerebral microvascular network [5], highlighting 104 the microvascular network as the primary system responsible for water inflow and out-flow in the brain (Figure 1).” However, CSF production in the choroid plexus is active secretion responsible for about 75-80 % of total CSF volume! The remaining CSF volume is produced by capillary filtration. The authors fail to mention that the modified Starling hypothesis is valid in the brain. In a healthy brain, capillary endothelium (intact blood-brain barrier) is the osmotic force that plays a major role in brain volume regulation.

The description of the glymphatic system is, unfortunately, confusing.

„Mechanisms of glioma-related edema formation” They state: „Peritumoral edema formation surrounding gliomas is considered to be driven by three main mechanisms [46–50]: (i) increased BBB permeability, (ii) tumor angiogenesis, 152 and (iii) increased aquaporin-4 expression (Figure 3).”

It is a logical error to separate increased BBB permeability from tumor angiogenesis. After all, the cause of the breakdown of the blood-brain barrier is often neovascularization, as the tumor develops fenestrated capillaries.

This statement that "Aquaporin 4 is expressed in astrocytes, which form the BBB, and is involved in the movement of water across the BBB" is entirely wrong. The role of AQP4 needs to be correctly described.

In its current form, this manuscript is not suitable for publication, on the one hand, due to the inaccurate description of the pathophysiology of edema formation and some factual errors.

Minor editing of English language required, however, I am not qualified to assess the quality of English in this paper.

Reviewer 3 Report

Well written narrative review, describing current understanding of brain edema formation in case of gliomas. One critical remark: Line 74 states that "Using CT, the edema surrounding the tumor appears as a region of low absorption and high intensity on T2-weighed images [19]". Yes, brain edema is also visible on CT, although is appears hypodense there. The paper of Pavliša G et al. [19] is about the magnetic resonance imaging, where edema is hyperintense. Consider replacing CT with MRI.

Some technical remarks:

Line 127 "internal cervical artery" should be "internal carotid artery".

Figure 2 and Figure 3 schematic numeration of the water drainage pathways is in arabic numerals (1,2,3) but in the explanation part - roman (i, ii, iii). Consider changing to the same type of numerals.

Reviewer 4 Report

In their paper entitled “Peritumoral edema in gliomas: a review of mechanisms and management”, the Authors discuss the clinical properties of peritumoral edema that normally associates with gliomas. In particular, they describe the mechanisms currently considered responsible for edema and discuss the therapies used to manage it.

The paper is of interest and suitable for Biomedicines. The state of art is well described, and many interesting and recent findings have been reported. Bibliography is up-to-date: 29 (out of 102: please, note that the list of references starts with two N.1) of the cited papers have been, indeed, published between 2019 and 2023. Moreover, the concepts reported are illustrated by explicative figures.

I only have a very minor comment: In both fig. 1 and Fig.2 the different pathways/mechanisms are indicated with numbers, while in the legends to the figures the same pathways/mechanisms are indicated with symbols like i) etc. Perhaps, for clarity, it should be better to use the same symbols in the figures and in the legends.

Round 2

Reviewer 1 Report

Ref: biomedicines-2560399
Title: Peritumoral edema in gliomas: a review of mechanisms and management
Journal: Biomedicines

The manuscript entitled: “Peritumoral edema in gliomas: a review of mechanisms and management” by Ohmura et al. is a well written article reviewing the clinical features and mechanisms of peritumoral edema in gliomas. The article has been greatly improved after the revisions made by the authors. However, I feel that there are still issues to be addressed before the final acceptance. Please find below some more minor corrections:

1) Still the authors do not explain briefly the WHO 2021 classification and the different grades according to this new classification.

2) Figure legends 2-3: please make a brief summary of what is written here and move the rest in the text of the manuscript.

3) MMPs do not melt the basement membrane but cleave -please correct and also please mention what they do in the extracellular matrix (ECM). 

Reviewer 2 Report

The correction has been made, and the manuscript can serve as a review.
